# Robo-advisor acceptance: Do gender and generation matter?

**Gianna Figà-Talamanca**[1]*, **Paola Musile Tanzi**[2], **Eleonora D'Urzo**[1]

**1** Department of Economics, University of Perugia, Perugia, Italy, **2** Department of Economics, University of Perugia, Perugia, Italy—Affiliate Professor SDA Bocconi, Milan, Italy

* gianna.figatalamanca@unipg.it

## Abstract

Robo-advice technology refers to services offered by a virtual financial advisor based on artificial intelligence. Research on the application of robo-advice technology already highlights the potential benefit in terms of financial inclusion. We analyze the process for adopting robo-advice through the technology acceptance model (TAM), focusing on a highly educated sample and exploring generational and gender differences. We find no significant gender difference in the causality links with adoption, although some structural differences still arise between male and female groups. Further, we find evidence that generational cohorts affect the path to future adoption of robo-advice technology. Indeed, the ease of use is the factor which triggers the adoption by Generation Z and Generation Y, whereas the perceived usefulness of robo-advice technology is the key factor driving Generation X+, who need to understand the ultimate purpose of a robo-advice technology tool before adopting it. Overall, the above findings may reflect that, while gender differences are wiped out in a highly educated population, generation effects still matter in the adoption of a robo-advice technology tool.

## 1. Introduction

A robo-advisor is a virtual financial advisor, based on artificial intelligence, that is currently in the spotlight in wealth management [1–4], propelling leading innovations in the financial industry. According to the European Securities and Markets Authority (ESMA) 2018 guidelines, "Robo-advice means the provision of investment advice or portfolio management services (in whole or in part) through an automated or semi-automated system used as a client-facing tool" [5].

According to Niszczota and Kaszás [6], the main obstacle to the adoption of robo-advice technology is the perception of humans as more effective than algorithms when the decision process requires subjective judgments. Rasiwala and Kohli [7], by contrast, evidence that robo-advice technology is gaining popularity, as it may reduce or eliminate the risk of human errors. Transparency, security and simplicity of processes are considered relevant factors for any information technology artifact and may lead to increased customer satisfaction and willingness to adopt robo-advice services specifically. The expectation of financial inclusion related to Fintech is high [8, 9], and Bianchi and Briere [10] confirm the potential of robo-advice

**Funding:** GFT Grant 2018.0427.021 Fondazione Cassa di Risparmio di Perugia (Italy) Grant FRB 2017-2019 Università di Perugia (Italy) PMT Grant

FRB 2017-2019 Università di Perugia (Italy) The
funders had no role in study design, data collection
and analysis, decision to publish, or preparation of
the manuscript.

**Competing interests:** The authors have declared
that no competing interests exist.

technology in this direction, suggesting that the human–robot interaction can improve financial capability. To the best of our knowledge, few researchers have analyzed factors that might influence the adoption of robo-advice by end users [6, 11–15]. Specifically, Hohenberger et al. [12] examine how financial experience influences the intention to use a robo-advisor and how emotional reactions, such as anxiety and joy, can mediate this effect. They find that individuals' behavioral intention increases when positive emotions are expected from using a robo-advisor but decreases otherwise; moreover, this relationship may change, according to individuals' self-enhancement motives (e.g. possibility of accumulating wealth).

Cheng et al. [14] evidence the significant influencing role of supervisory control and validate the relationships between trust influencing factors: trust in technologies, trust in vendor and trust in a robo-advisor.

Finally, Lourenço et al. [15] focus on the impact of firm characteristics on consumer acceptance of pension investment advice generated by a robo-advisor. They find that consumers' perceptions of trust and expertise of the firm providing the automated advice are important drivers of advice acceptance. Hohenberger et al. [12], Cheng et al. [14] and Lorenço et al. [15] measure the impact of the proposed adoption drivers through a structural equation model (SEM), in line with the technology acceptance methodology (TAM) by Davis [16]. Indeed, the TAM is possibly the most widely used framework in the field of information systems for measuring and forecasting technology adoption [16] and has proved empirically useful in several studies [17, 18].

With respect to technology adoption, several studies in many fields have documented the relevance of generational cohorts and gender (e.g. [19–24]). Generational cohorts have a different culture and approach to technology in general, and this effect can be exacerbated when technology is related to possible profits and losses. Further, women are often proved to be less self-confident (see, among others, [25]), and this may impact their attitude. In this regard, Chen et al. [24] find a significant gender gap in the use of Fintech services.

The impact of generational cohorts/age and gender for robo-advisory application is investigated in D'Acunto et al. [26], Lourenço et al. [15] and Cheng et al. [24], with partially conflicting results. D'Acunto et al. [26] evidence no demographic differences between users and non-users of robo-advice; Cheng et al. [14] include gender and age groups among control variables to specifically investigate trust transfer in robo-advice and show that they have insignificant effects in their research model. Conversely, Lourenço et al. [15] evidence that age has a negative and significant influence, such that older individuals and women are less satisfied with using automated tools than younger individuals. Interestingly, Lourenço et al. [15] also evidence a positive impact of education level on robo-advice acceptance.

We enter the current debate by investigating the moderation effect of gender and generational cohorts on the TAM model, applied to the robo-advisor technology, by focusing on a homogeneous, highly educated sample, to avoid possible biases related to investor education level. Indeed, generational and gender characteristics may have different effects in a highly educated framework.

The main question we address in this paper is whether there is a significant difference in the factors affecting the acceptance of the robo-advice technology across the selected generational groups and between genders. Specifically, we survey students and employees (faculty and staff) of the University of Perugia, Italy; all respondents are of working age. Hence, elders are not included; thus, the effect of biological aging is beyond the scope of this analysis.

The paper is organized as follows: section 2 describes the research methods and states our hypotheses, section 3 describes how data are collected and treated and section 4 summarizes the empirical results. Finally, section 5 discusses the implications of the research outcomes and offers some concluding remarks.

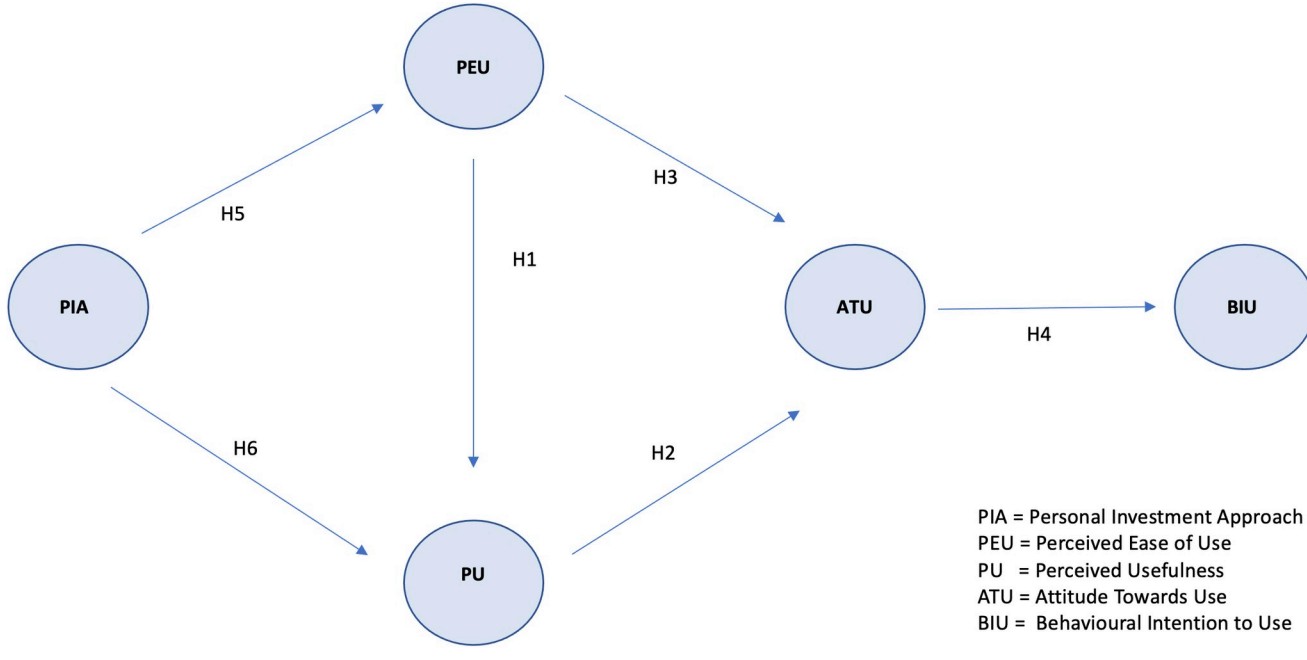

**Fig 1. The technology acceptance model.**

## 2. Research methods and hypotheses

The TAM [16] is a path model that links specific constructs (latent variables) through causal relationships to be estimated in a SEM setting. According to the original TAM model, represented in Fig 1, the variables linked to the acceptance of any technology and its adoption are *perceived ease of use* (PEU), *perceived usefulness* (PU), *attitude towards use* (ATU) and *behavioral intention to use* (BIU).

Davis [16] defines *perceived usefulness* as "the prospective user's subjective probability that using a specific application system will increase his or her job performance within an organizational context" and *perceived ease of use* as "the degree to which the prospective user expects the target system to be free of effort."

According to the TAM, *behavioral intention to use* refers to the prospective use of a given information system and thus determines technology acceptance. *Attitude towards use* and *perceived usefulness* jointly influence *behavioral intention*, which is indirectly affected by *perceived ease of use* as well. *Attitude towards use* is directly affected by both *perceived usefulness* and *perceived ease of use*, and *perceived usefulness* is directly influenced by *perceived ease of use*. Further, according to the TAM, *perceived usefulness* and *perceived ease of use* may be affected by external variables (EXT). In turn, both *perceived usefulness* and *perceived ease of use* mediate the effect of external variables on the user's *attitude towards use* and *behavioral intention to use* (Fig 1).

The purpose of this paper is to apply the TAM to the adoption of a robo-advisor service, by analyzing whether the propensity to use this technology varies across gender or generational groups. In this study, *perceived ease of use* is defined as the degree to which a respondent believes that learning to use a robo-advisor requires a relatively low degree of effort, and *perceived usefulness* is defined as the degree to which a respondent believes that using a robo-advisor would enhance their investment performance. The definition of the other variables is intuitive: *attitude* is the individual's positive or negative feelings about a behavior, and

*behavioral intention* is the individual's perceived probability that they will use the system, here the robo-advisor.

Moreover, we introduce a new construct, the *personal investment approach* (PIA), to represent the propensity of individuals to actively manage their portfolio, and we assume that this construct also influences the technology acceptance of robo-advice as an external variable (EXT = PIA).

Specifically, we test whether and to what extent the TAM links are moderated by generational cohorts or by gender.

Moderation analysis can be performed alternatively by using interaction terms or by splitting the sample into sub-groups of interest before estimation (so-called multi-group analysis [MGA]). We adopt the latter approach, which permits assessing both the magnitude of TAM path relationships within sub-groups and the significance of their differences. The hypotheses tested in the paper are summed in Table 1.

## 3. Data collection, measures and procedure

We designed an online questionnaire using SurveyMonkey (https://www.surveymonkey.com/) and submitted the questionnaire link to students, faculty and staff of the University of Perugia in October 2019. Overall, 214 respondents completed the questionnaire and are considered a convenience sample for the following analysis. The response rate for the complete questionnaire is around 5% for University faculty and staff and 0.45% for University students (around 25,000 enrolled in 2019/2020). Section 1 of the questionnaire (12 questions) allows us to establish the sample's descriptive characteristics, summarized in Table 2.

The sample is composed of highly educated individuals. Most respondents hold at least a bachelor's degree, of whom 20% also have a post-graduate diploma. Further, those having only a high school degree are mainly students currently pursuing their bachelor's. Overall, the sample fits well the aim and context of this study.

Only one respondent does not disclose information about gender; 63% are females and 37% are males. Respondent ages range from 18 to 67 years, grouped by generational cohort:

**Table 1. Hypotheses under investigation.**

| H1 | | The *perceived ease of use* of the robo-advisor positively affects its *perceived usefulness* |
|---|---|---|
| | a. | The effect is moderated by the generation cohort |
| | b. | The effect is moderated by the gender |
| H2 | | The *perceived usefulness of* the robo-advisor positively affects the *attitude towards its use*. |
| | a. | The effect is moderated by the generation cohort |
| | b. | The effect is moderated by the gender |
| H3 | | The *perceived ease of use* of the robo-advisor positively affects the *attitude towards its use*. |
| | a. | The effect is moderated by the generation cohort |
| | b. | The effect is moderated by the gender |
| H4 | | The *attitude towards use* of the robo-advisor positively affects the *behavioral intention* of its adoption. |
| | a. | The effect is moderated by the generation cohort |
| | b. | The effect is moderated by the gender |
| H5 | | The *personal investment approach* positively affects the *perceived ease of use* of the robo-advisor. |
| | a. | The effect is moderated by the generation cohort |
| | b. | The effect is moderated by the gender |
| H6 | | The *personal investment approach* positively affects the *perceived usefulness* of the robo-advisor. |
| | a. | The effect is moderated by the generation cohort |
| | b. | The effect is moderated by the gender |

**Table 2. Questionnaire: Section 1, descriptive statistics.**

| Attributes | Frequency (N = 214) | Percentage |
|---|---|---|
| Gender | | |
| Male | 79 | 36.9 |
| Female | 134 | 62.6 |
| Other | 1 | 0.5 |
| Generational group | | |
| Generation Z | 102 | 47.7 |
| Generation Y | 42 | 19.6 |
| Generation X[+] | 70 | 32.7 |
| Education | | |
| High school | 90 | 42.1 |
| Bachelor's degree | 81 | 37.9 |
| Graduate studies | 43 | 20.1 |
| Occupation | | |
| Student | 102 | 47.7 |
| Other | 112 | 52.3 |
| Financial literacy | | |
| No | 85 | 39.7 |
| Intermediate | 119 | 55.6 |
| High | 10 | 4.7 |
| Confidence in the evaluation of financial offers | | |
| No, I trust the proposal | 34 | 15.9 |
| No, I am not confident but I try to obtain external information | 135 | 63.1 |
| Yes, I am usually able to evaluate | 45 | 21.0 |
| Use of internet-based information | | |
| Often | 101 | 47.2 |
| Seldom | 63 | 29.4 |
| Never | 50 | 23.4 |
| Investment horizon (years) | | |
| <1 | 88 | 41.1 |
| 3 | 49 | 22.9 |
| 5 | 45 | 21.0 |
| 10 | 20 | 9.3 |
| >10 | 12 | 5.6 |
| Trusted advisors support | | |
| Yes | 111 | 51.9 |
| No | 101 | 47.2 |
| Previous use of robo-advice | | |
| No | 210 | 98.1 |
| Yes | 4 | 1.9 |
| Interest in robo-advice | | |
| No | 142 | 66.4 |
| Yes | 72 | 33.6 |
| Cyber-risk aversion | | |
| No, I trust | 16 | 7.5 |
| Yes, that's why I am not interested | 79 | 36.9 |
| Yes, but I would use it anyway | 35 | 16.4 |
| I have no idea | 84 | 39.3 |

Generation Z (born 1994–2012; 47%), Y (born 1980–93; 19.6%) and X$^+$ (born before 1980; 32.7%). A generational cohort is a group of individuals identified by common social and historical life events [27]; however, the definition is usually loose and has been questioned by Rudolph and Zacher [28]. For the purposes of proper contextualization with mainstream literature, we cautiously use generational groupings consistent with those defined in [21, 29, 30] while acknowledging the theoretical limitation highlighted in [28].

After asking for basic demographic information, the survey investigates respondents' level of financial literacy and awareness, familiarity with the internet and willingness to use innovative technologies such as robo-advice.

With respect to financial literacy, most respondents (60.3%) report having an intermediate or high level of financial education. Notably, only 16% declare a *low ability* to evaluate investment proposals, and 21% declare a *high ability* to evaluate investment choices, in line with the overconfidence of investors evidenced in behavioral finance studies (e.g. [31–33]). In addition, nearly half the respondents report using the internet often to find information necessary to making financial decisions. Although most have a long life expectancy, they plan their investments within a very short time horizon (less than 1 year for 41.1% of respondents), losing the opportunities offered by true investment planning, which usually takes advantage of a longer investment horizon.

With respect to the use of robo-advice, only 2% declare having previously used such technology, as expected. Moreover, only a minority are interested in future use of robo-advice, in line with other findings [34]. Not surprisingly, of the 72 respondents declaring interest in this technology, 58.3% belong to Generation Z. In addition, interested respondents declare that they would use robo-advice only for small amounts (45.1%), for low-fee services (29.6%) or on well-known investment platforms (22.5%).

The low interest in robo-advice is partially motivated by respondents' cyber-risk aversion: only 7.5% fully trust algorithms and internet-based platforms, and only 16.4% show enough risk tolerance to pursue using internet-based solutions.

Note that the proportion of respondents interested in the future application of robo-advice is not conditioned by their previous reliance on a trusted professional (interested: all sample = 33.6%; interested: no advisor = 33.7%; interested: with advisor = 32.4%).

The second and core part of the questionnaire includes five questions with 7-point Likert-scale responses to measure the TAM constructs used in this study. Likert-scale levels indicate respondents' agreement or disagreement with the proposed assertions (1 = *Strongly agree*; 7 = *Strongly disagree*). The complete questionnaire is reported in Appendix A in S1 Appendix.

Before estimating the TAM links, we assess the validity of the measurement model through suitable statistics such as composite reliability (CR) and average variance extracted (AVE), as recommended in [35, 36], among others. Indeed, latent constructs are sufficiently reliable and display a quite satisfactory AVE, slightly below the desired level for the PIA variable.

In addition, no collinearity issues are evidenced, since the variance inflation factor (VIF) is far below the conservative 3.00 threshold for all latent constructs and across sub-groups.

Following Hanseler et al. [37] and Mahmoud et al. [21], among others, we also apply the measurement invariance of the composite model (MICOM) procedure to the TAM constructs. The outcomes of a permutation test (1,000 permutations) show that the original correlations of the constructs, when segmented by gender or generational cohort, are not statistically different from 1, and that the null hypothesis of compositional invariance cannot be rejected with all p-values far above 5%. Details of this model validation assessment are summarized in Appendix B in S1 Appendix.

## 4. Empirical results

As a preliminary step, the TAM is estimated on the entire dataset, evidencing positive and significant links between the TAM latent constructs. A full description of this preliminary analysis is provided in Appendix C in S1 Appendix. In particular, we find that model outcomes do not change significantly in the whole sample by controlling for gender, whereas a moderation effect is evidenced for generational groups.

This finding further motivates the MGA analysis to investigate which and to what extent causality links differ between genders and between generational groups. Specifically, the parameter estimates are obtained via the Multi-Group SmartPLS routines [38], and the outcomes are illustrated in Figs 2 and 3, respectively, for the generational and gender groups. Path coefficients are reported for each of the hypotheses together with the p-value (in brackets) for the corresponding significance t-test.

Hypothesis H1 holds true across generational and gender groups: perceived ease of use (PEU) of robo-advice has a positive and significant impact on perceived usefulness (PU). Specifically, the path coefficients are 0.566, 0.587 and 0.263 respectively for Generations Z, Y and X+ and are strongly significant ($p < 0.001$) for the first two groups. As for gender, perceived ease of use (PEU) has a strongly significant impact on perceived usefulness (PU), with path coefficients of 0.448 and 0.424 respectively for male and female respondents.

Similarly, H2 is fully confirmed: the path coefficients from perceived usefulness (PU) to attitude towards use (ATU) are very high across all generation groups (0.560, 0.690, 0.743) and genders (0.821 and 0.635) and are strongly significant ($p < 0.001$).

The relationship in H3 between *perceived ease of use* (PEU) and *attitude towards use* (ATU) holds true for Generation Z, with a strongly significant path coefficient (0.298, $p = 0.001$); it is lower (0.229) and weakly significant ($p = 0.078$) for Generation Y and vanishes (0.067) for

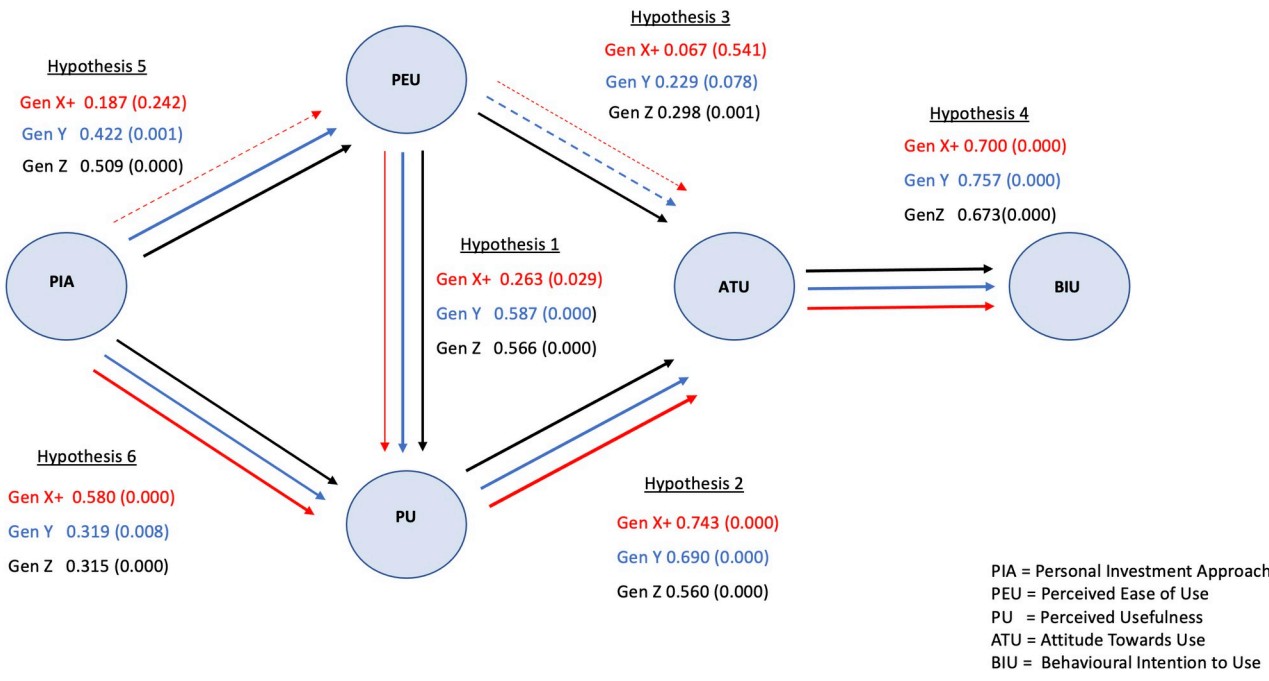

For each hypothesis the path coefficient estimate is reported with the p-value (in brackets) of the t-test.

**Fig 2. Technology acceptance constructs and causality links segmented by generation.**

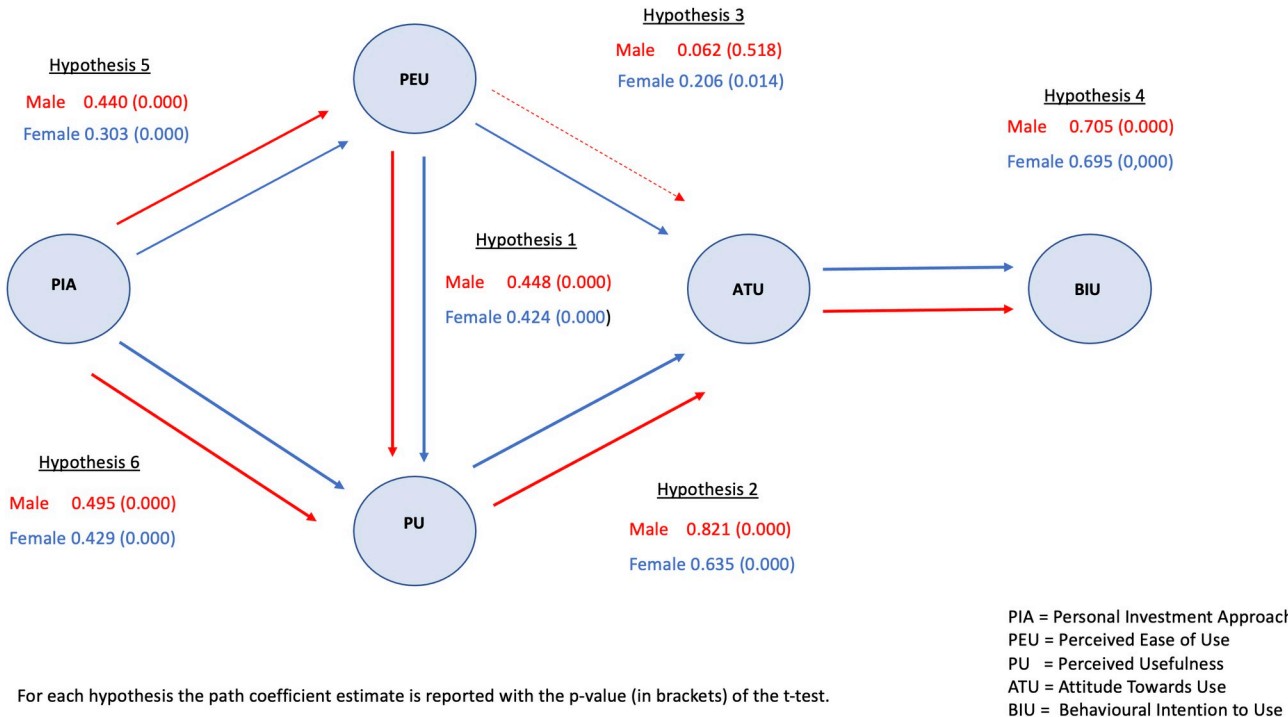

For each hypothesis the path coefficient estimate is reported with the p-value (in brackets) of the t-test.

**Fig 3. Technology acceptance constructs and causality links segmented by gender.**

Generation X$^{+}$ (p = 0.541). Further, the link is positive and significant for the female group (0.206 with a p-value of 0.014), but non-significant for the male group, evidencing that perceived ease of use may act as a driver in the adoption of a robo-advisor for Generation Z and females.

Hypothesis H4 is valid for all considered groups: attitude towards use (ATU) is positively linked to behavioral intention (BIU), with path coefficients of 0.673, 0.757 and 0.700 respectively for the three generational cohorts, and 0.705 and 0.695 for males and females. All parameters are strongly significant, with p-values lower than 0.001.

The effect in H5 of personal investment approach (PIA) on perceived ease of use (PEU) is strongly significant, except for Generation X$^{+}$. Indeed, the magnitude of the parameter decreases across generational cohorts, with path coefficients of 0.509, 0.422 and 0.187 respectively for Generations Z, Y and X$^{+}$. Links are positive and significant for both genders, with slightly higher values for the male group.

Finally, the impact of *personal investment approach* (PIA) on *perceived usefulness* (PU) H6 is confirmed to be positive for all generational groups (path coefficients of 0.315, 0.319 and 0.580) and both genders (path coefficients of 0.495 and 0.429). The results are strongly significant (p < 0.001) across all groups.

The Multi-Group SmartPLS routine includes statistical tests, based on the bootstrap algorithm provided by SmartPLS, to test whether the above differences between parameter estimates of different groups are significant, and hence to assess the presence and the magnitude of moderation effects. The outcomes of these tests are summarized in Table 3 for generational and gender groups; both the relative differences of the path coefficients and the corresponding statistical test p-values are displayed. Notably, no significant difference is detected for any path coefficient between Generation Y and Generation Z. Conversely, some causality links are significantly different between Generation X$^{+}$ and the two other groups. Specifically, the effect of

**Table 3. Outcomes of the tests between generational cohorts and gender groups.**

| Hypothesis | | Gen Y vs Gen Z | | Gen X + vs Gen Z | | Gen X+ vs Gen Y | | Females vs Males | |
|---|---|---|---|---|---|---|---|---|---|
| | | Rel. diff. | p-value | Rel. Diff. | p-value | Rel. Diff. | p-value | Rel. Diff. | p-value |
| H1 | PEU → PU | 0.022 | 0.831 | −0.303 | 0.014 | -0.324 | 0.041 | −0.024 | 0.844 |
| H2 | PU → ATU | 0.130 | 0.375 | 0.184 | 0.153 | 0.053 | 0.718 | −0.186 | 0.090 |
| H3 | PEU → ATU | −0.069 | 0.656 | −0.231 | 0.109 | -0.162 | 0.347 | 0.144 | 0.259 |
| H4 | ATU → BIU | 0.084 | 0.339 | 0.027 | 0.736 | -0.058 | 0.584 | -0.010 | 0.875 |
| H5 | PIA → PEU | −0.088 | 0.563 | −0.332 | 0.041 | -0.234 | 0.228 | −0.137 | 0.297 |
| H6 | PIA → PU | −0.004 | 0.957 | 0.265 | 0.033 | 0.261 | 0.078 | −0.066 | 0.555 |

Note. ATU = attitude towards use; BIU = behavioral intention to use; PEU = perceived ease of use; PIA = personal investment approach; PU = perceived usefulness.

*perceived ease of use* on *perceived usefulness* is significantly lower for Generation X$^+$, confirming the moderation effect of generation (H1a). In addition, the magnitude of the effect of *personal investment approach* (PIA) on *perceived ease of use* (PEU) is lower for Generation X$^+$ than for either Generation Y or Generation Z, but only the latter is significant. Conversely, the effect of *personal investment approach* (PIA) on *perceived usefulness* (PU) is higher for Generation X$^+$ with respect to the other two groups. Once again, the difference between Generations X$^+$ and Z is significant, whereas the difference between Generations X$^+$ and Y is only weakly significant.

Summing up, the outcomes in Table 3 evidence that causal relationships in H1, H5 and H6 are moderated by generational cohort, with a significant difference between Generations X$^+$ and Z for H1, H5 and H6 and a significant difference between Generations X$^+$ and Y for H1. Hence, only H1a, H5a and H6a hold true.

Interestingly, the path coefficients are not significantly different between the two gender groups, as reported in Table 3, except for the causality link from *perceived usefulness* to *attitude towards use*, for which the difference is negative and weakly significant. Hence, only H2b is weakly validated. Note also that although non-significant, the difference between the causality link *perceived ease of use* to *attitude towards use* is positive for the female and male groups and negative between the younger and older generation groups.

## 5. Discussion and concluding remarks

We apply the TAM to assess whether the causality links for robo-advisory adoption change significantly between generational groups and genders when focusing on a highly educated sample.

Results evidence that the strongest path towards the intention to use a robo-advisor is given by the link-chain *perceived usefulness → attitude towards use → behavioral intention*, and that the magnitude of the effect increases from Generation Z to Generation X$^+$, evidencing how *usefulness* is the main driver for adoption in the latter case. Notably, the direct link from *perceived ease of use* and *attitude towards use* is significant only for Generation Z, which appears to be the only group driven by *ease of use* of the technology. The analysis also suggests a positive and significant inner link from *perceived ease of use* to *perceived usefulness* for all generational groups. The effect is stronger for Generations Z and Y and appears to be triggered by their *perceived ease of use* of robo-advisory.

As for the effect of the *personal investment approach*, we find interesting outcomes: the link between this external factor and the *perceived ease of use* of robo-advice is irrelevant for Generation X$^+$ and significant for the other groups, with the highest value for Generation Z. We attribute this behavior to the fact that those young adults who actively manage their investment

portfolios are most likely already acquainted with online trading platforms and find no hurdle in the use of robo-advice.

Conversely, the link from *personal investment approach* to *perceived usefulness* is strongly significant for all generations, with a large impact for Generation X[+]; indeed, a higher propensity in mature adults to actively manage the investment portfolio may increase the perception of the robo-advisory as a useful tool.

Looking at the whole picture (Fig 2) shows that the causality path to adoption of a robo-advisor is quite different for the three generational groups. Indeed, if we follow the thickest links in the graph, we find that Generations Z and Y are both driven towards adoption by the *perceived ease of use* construct. However, while the chain-link for Generation Y goes through the potential usefulness of the technology, Generation Z may be significantly driven towards adoption directly from ease of use, without considering its usefulness. Indeed, within this group, both the links from *perceived ease of use* to *attitude towards use* and to *perceived usefulness* are positive and strongly significant.

A different path sequence is highlighted for Generation X[+]. Interestingly, this generation moves towards adoption by completely ignoring the ease of use of the technology and paying attention to its usefulness only. Ease of use appears to worth less than usefulness, a finding perhaps explained by the greater experience and maturity of this generation compared with the other two.

Concerning gender, the highest path coefficient for males is related to the path from *perceived usefulness* to *attitude towards use*, whereas for females it corresponds to the path from *attitude towards use* to *behavioral intention*. Moreover, females seem to be partially driven to *attitude towards use* by *perceived ease of use*, a link which is instead irrelevant for males. However, the differences between path coefficients are non-significant for all the causality links, evidencing that gender does not moderate the adoption of robo-advisory within a highly educated sample. Though it may come as a surprising result, a clue in this direction could have been the proportion of female respondents, showing the interest of highly educated women on the topic of the survey.

Overall, the above findings may reflect that whereas gender differences are zeroed out in a highly educated population, generation effects still matter in the adoption of robo-advice. As a matter of fact, an intrinsic limitation of this study and those taking into account behavioral differences in generational cohorts, is the possibility that such variations may be also generated by other characteristics, such as age or status, of the corresponding groups [28]. Such a circumstance can be verified only by future studies. In our setting, one should analyze the behavior of generation groups towards new technology instruments supporting financial decisions in the next 10 to 30 years to disentangle whether their approach is triggered by age rather than by cohort [39].

Another limitation of our research is the small sample size, based on students, staff, and faculty of the University of Perugia (Italy), which is not perfectly balanced with respect to gender and generational groups and makes the results not widely generalizable.

Further, as usual in this kind of surveys, the behavioral intention can influence the answers to the questions measuring the antecedent variables, leading to a potential risk of endogeneity. However, our hope is that our research could still catch a positive signal for future generations, where gender is not relevant, facing financial innovation technology, starting from an equal education. Besides, our results may help stimulate the discussion on robo-advising. Indeed, to contribute the diffusion of such technology, providers should highlight the potential usefulness of the technology to mature customers. By contrast, our results suggest some concerns, which may be relevant for regulatory authorities, on the risks for Generation Z to adopt such technology just driven by its ease of use.

## Supporting information

**S1 Appendix.**
(DOCX)

**S1 Data.**
(XLSX)

## Author Contributions

**Conceptualization:** Gianna Figà-Talamanca, Paola Musile Tanzi, Eleonora D'Urzo.

**Data curation:** Gianna Figà-Talamanca, Paola Musile Tanzi, Eleonora D'Urzo.

**Formal analysis:** Gianna Figà-Talamanca, Paola Musile Tanzi, Eleonora D'Urzo.

**Funding acquisition:** Gianna Figà-Talamanca, Paola Musile Tanzi.

**Investigation:** Gianna Figà-Talamanca, Paola Musile Tanzi.

**Methodology:** Gianna Figà-Talamanca, Eleonora D'Urzo.

**Software:** Gianna Figà-Talamanca, Eleonora D'Urzo.

**Supervision:** Gianna Figà-Talamanca, Paola Musile Tanzi.

**Writing – original draft:** Gianna Figà-Talamanca, Paola Musile Tanzi, Eleonora D'Urzo.

**Writing – review & editing:** Gianna Figà-Talamanca, Paola Musile Tanzi.

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
