## [Decision Letter · Decision Letter 0]

2 Jul 2021

PONE-D-21-14526

How does investor generational cohort make a difference in the acceptance of financial robo-advice?

PLOS ONE

Dear Dr. FIGA'-TALAMANCA,

Thank you for submitting your manuscript to PLOS ONE. After careful consideration, we feel that it has merit but does not fully meet PLOS ONE’s publication criteria as it currently stands. Therefore, we invite you to submit a revised version of the manuscript that addresses the points raised during the review process.

The manuscript would benefit from better engagement with relevant recent research on generational differences [1]. Further, please, follow JF Hair, JJ Risher, M Sarstedt and CM Ringle [2] guidelines to perform and report PLS-SEM results, especially in terms of the “Measurement Invariance of the Composite Models” (MICOM) procedure.

References

1. Mahmoud AB, Hack-Polay D, Grigoriou N, Mohr I, Fuxman L: **A generational investigation and sentiment and emotion analyses of female fashion brand users on Instagram in Sub-Saharan Africa**. *Journal of Brand Management *2021.

2. Hair JF, Risher JJ, Sarstedt M, Ringle CM: **When to use and how to report the results of PLS-SEM**. *European Business Review *2019, **31**(1):2-24.

We look forward to receiving your revised manuscript.

Kind regards,

Ali B. Mahmoud, Ph.D.

Academic Editor

PLOS ONE

Journal Requirements:

3. Please consider changing the title so as to meet our title format requirement (https://journals.plos.org/plosone/s/submission-guidelines). In particular, the title should be "Specific, descriptive, concise, and comprehensible to readers outside the field" and in this case it is not informative and specific about your study's scope and methodology.

Reviewers' comments:

Reviewer's Responses to Questions

**Comments to the Author**

1. Is the manuscript technically sound, and do the data support the conclusions?

Reviewer #1: Yes

Reviewer #2: Partly

2. Has the statistical analysis been performed appropriately and rigorously? 

Reviewer #1: Yes

Reviewer #2: I Don't Know

3. Have the authors made all data underlying the findings in their manuscript fully available?

Reviewer #1: Yes

Reviewer #2: Yes

4. Is the manuscript presented in an intelligible fashion and written in standard English?

Reviewer #1: Yes

Reviewer #2: Yes

5. Review Comments to the Author

Reviewer #1: Dear author! Thank you for submitting your manuscript. It is an interesting topic with some interesting findings. On the other hand, the manuscript has some issues. Below these issues are enclosed.

1) The first key issue is that the manuscript does not reasonably cover the literature on the context of Robo-advisor. One example is:

Cheng, X., Guo, F., Chen, J., Li, K., Zhang, Y., & Gao, P. (2019). Exploring the trust influencing mechanism of robo-advisor service: a mixed method approach. Sustainability, 11(18), 4917.

2)The second key issue is section number 4 (Discussion and concluding remarks). The manuscript should discuss the results (not only elaborate on the result). Several questions should be highlighted and recent related studies need to be highlighted. Moreover, the theoretical contribution should be more precise.

3) It is not clear why it is important to compare the “Digital Natives, Millennials and previous generations.”

The two studies, the manuscript, utilizes (Fulk et al., 2018; D'Acunto, 2019) did not find a significant difference in terms of age, Hence, it is critical to explain why it is important to do this comparison.

4) The manuscript argues that “Generation Y, or Millennials, which includes people born between 1980 and the early 2000s, are digitally advanced and potentially open to technology innovation in the financial sector, such as robo-advisors; however, one may argue that complete acceptance of such disruptive innovation would be achieved only by the younger Digital Natives”. It is difficult to know if this represents a subjective opinion or an assumption. Taken into consideration that the manuscript reported that “Notably, Digital Natives and Millennials show a very similar pattern, and no significant difference is detected between path coefficients.”

It is worth pointing out that several studies (e.g. Gomber et al., 2018) have reported that a high percentage of Millennials use FinTech business solutions. The manuscript should support the above argument with recent numbers, surveys, or/and studies.

Gomber, P., Kauffman, R. J., Parker, C., & Weber, B. W. (2018). On the fintech revolution: Interpreting the forces of innovation, disruption, and transformation in financial services. Journal of Management Information Systems, 35(1), 220-265.

5)It is not clear why it is important to study the opinion of the educated respondents. This also arises an enquiry regarding the possibility to generalize the results of the study.

6)The construct ‘Personal Investment Approach’ the manuscript used, represents a nice contribution. However, the manuscript needs to report the reliability and validity measures for this construct.

7)The Abbreviation ESMA needs to be clarified. There is an opportunity that some readers might not be familiar with the abbreviation of European Securities and Markets Authority

Reviewer #2: This paper investigates the moderating role of age and the differences across age groups on the technology acceptance model (TAM) in the context of financial robo-advice.

The authors find that age, recoded into 'generational cohorts', moderates some of the relationships of the TAM, and that there are differences across some of the age groups. In general, the paper is well-written. I have a few remarks and concerns regarding the paper.

RESEARCH QUESTIONS AND CONCEPTUALIZATION

1. The motivation and relevance of the research questions (the moderating effect of age) should be strengthened. Currently, the authors mention they "believe" age may be important.

While the (moderating) effects of age have been investigated in several technology-related domains before, why is age important in the specific context of robo-advice, in particular financial robo-advice? Lourenço, Dellaert, and Donkers (2020), for instance, have included age as a control in their recent study of robo-advice acceptance.

2. And then the authors discuss why there could be differences between the two youngest groups but not between those and the oldest groups.

3. The authors should also discuss, even if briefly, the differences between age-, cohort-, and period-effects and the challenges to disentangle the three. In so doing, they should clarify that they recode age into generational cohorts out of convenience, but they do not intend estimate the effects of age vs. cohort (or age vs. period nor cohort vs. period).

4. The hypotheses, which could perhaps be re-located to section 1, should be formulated in reference to the moderating effect of age and the differences across cohorts. The current hypotheses refer to the original TAM model, which have been tested countless times before.

5. In the introduction, the authors could add on p. 3, line 34 the work by Baker and Dellaert (2017).

6. While the authors also emphasize, and analyze in detail, the effect of gender, the introduction and conceptualization are silent about it. As such, mentioning on page 6, lines 98-100, that the purpose of the study is two-fold comes as a surprise, to say the least.

METHODOLOGY AND RESULTS

1. The authors do not explain how they have recoded age into generational cohorts, and what past sources have they followed to do that. That is, what birth years (current age) were considered to classify individuals into the four different cohorts?

2. The authors should also be consistent in their use of generational labels. For instance, sometimes they refer to young individuals as Millennials and sometimes as Generation Z, or Digital Natives and Generation Y. Also, are Millennials Gen. Z or Gen. Y? More importantly, 'Digital Natives' may be misleading, as Millennials are often considered Digital Natives too.

3. The authors should report the fit measures typically used when estimating structural equation models, and show in particular that a model with age (and age cohorts) outperforms in terms of fit a model without age. They could use a likelihood ratio test.

4. Do the results change if instead of a multigroup analysis the authors estimate a model with the entire sample but include the moderating variables (3 dummy variables, one for each of the three cohorts that compare to the reference cohort, Gen. Z)? While comparing such model, the authors should not only incude the moderating variables but also the cohort dummies as main effects.

5. What is the number of respondents who fall into the two oldest cohorts, Gen. X and Baby Boomers? In Table 1, we can only see that, jointly, there are 70 respondents in the 'other' age group.

6. What is the response rate, i.e. the ratio of respondents to the number of individuals who received the link?

CONCLUSIONS

The authors should discuss, even if briefly, the limitations of their study, namely (i) the fact that it is only based on self-reported measures (it is not a behavioral study where respondents get to evaluate an actual financial robo-advice algorithm or interface), and (ii) their sample is an Italian sample of highly educated individuals.

Typographical errors and other writing errors to correct:

p.3, line 27: 'it's' should be 'it is';

p. 4, line 60: there seems to be something missing, perhaps 'also on' right after 'but';

p. 4, line 72: should be 'four' instead of 'three'

p. 5, line 88: should be 'prospective' instead of 'actual';

p. 10, line 155: 'it's' should be 'it is';

p. 13, lines 220-231: use 'to' instead of arrows;

In the Figures, the first variable should be 'PIA' instead of 'EXT';

References used in this report:

Baker, Tom, and Benedict Dellaert. "Regulating robo advice across the financial services industry." Iowa L. Rev. 103 (2017): 713.

Lourenço, C. J. S., Dellaert, B. G. C., & Donkers, B. (2020). Whose algorithm says so: The relationships between type of firm, perceptions of trust and expertise, and the acceptance of financial robo-advice. Journal of Interactive Marketing, 49, 107–124.

6. PLOS authors have the option to publish the peer review history of their article (what does this mean?). If published, this will include your full peer review and any attached files.

Reviewer #1: No

Reviewer #2: No

---

## [Author Response · Author response to Decision Letter 0]

22 Oct 2021

I have attached three reports with detailed answers to the Academic Editor and the two reviewers, respectively. Please, see attached files.

---

## [Decision Letter · Decision Letter 1]

15 Dec 2021

PONE-D-21-14526R1Robo-advisor acceptance: do gender and generation matter?PLOS ONE

Dear Dr. FIGA'-TALAMANCA,

Thank you for submitting your manuscript to PLOS ONE. After careful consideration, we feel that it has merit but does not fully meet PLOS ONE’s publication criteria as it currently stands. Therefore, we invite you to submit a revised version of the manuscript that addresses the points raised during the review process.

We look forward to receiving your revised manuscript.

Kind regards,

Ali B. Mahmoud, Ph.D.

Academic Editor

PLOS ONE

Reviewers' comments:

Reviewer's Responses to Questions

**Comments to the Author**

1. If the authors have adequately addressed your comments raised in a previous round of review and you feel that this manuscript is now acceptable for publication, you may indicate that here to bypass the “Comments to the Author” section, enter your conflict of interest statement in the “Confidential to Editor” section, and submit your "Accept" recommendation.

Reviewer #2: (No Response)

Reviewer #3: (No Response)

2. Is the manuscript technically sound, and do the data support the conclusions?

Reviewer #2: Partly

Reviewer #3: Yes

3. Has the statistical analysis been performed appropriately and rigorously? 

Reviewer #2: I Don't Know

Reviewer #3: Yes

4. Have the authors made all data underlying the findings in their manuscript fully available?

Reviewer #2: Yes

Reviewer #3: Yes

5. Is the manuscript presented in an intelligible fashion and written in standard English?

Reviewer #2: No

Reviewer #3: Yes

6. Review Comments to the Author

Reviewer #2: This paper’s topic (acceptance of financial robo-advice) is a timely one. And though demographic measurements are typically used for profiling they may hold some predictive or even explanatory power of their own, relevant for practitioners and policy makers. In that sense, I understand why one could be interested in the moderating effect of age — and gender.

However, the paper has several limitations. I hope the following list helps the authors strengthen their study and the manuscript.

RESEARCH QUESTIONS AND CONCEPTUALIZATION

1. The motivation, managerial and academic relevance, and potential contribution of the study’s main research question — the moderating role of age and gender on the effect of the TAM antecedents on the intentions to use a financial robo-advisor — are again not sufficiently thought through or discussed.

2. Specifically, why are age and gender important to study, managerially and academically, in the context of financial robo-advice adoption, and given what we already know about technology acceptance in general?

3. In the Introduction, there is only one paragraph where age and gender are mentioned, and it is stated there that “several studies have documented the relevance of age and gender”, although the authors do state that in a few studies the results are “conflicting”. Is this list exhaustive? What is the direction of the effects of age and gender if one would enlarge the literature review on those variables? Searching for “technology acceptance by older adults” or “by older people” and related terms shows a long list of studies and review papers.

4. More importantly, how exactly would we expect age and gender to have a moderating effect in the context at hand?

5. One could very well imagine that age makes individuals outdated in terms of technology awareness and have a hard time using a new technology, among other effects, as could certainly be predicted from the TAM model and has been shown before. This prediction, or a similar one, though not new, is never discussed.

6. But one could further imagine that there is theory and evidence to support a moderating effect of age in the context of financial robo-advice, possibly different from the one regarding technology acceptance in general. For instance, if older individuals become not only more interested in financial products due to being close to retirement but also maybe more comfortable in admitting to an algorithm rather than a human advisor their limited financial literacy.

7. For gender, one could very well imagine that the higher risk aversion that has been documented among females in many studies could be at play here, and help generating hypotheses — or leaving the matter for empirical scrutiny in case there would be conflicting arguments when it comes to financial robo-advice in particular (maybe the higher risk aversion among females only shows in interaction with education, which would mean that the effect would not show in a highly educated sample).

8. But no such arguments, or similar ones, are discussed in the paper.

9. In the “Research methods and hypotheses” section (by the way, it would be better to avoid discussing methodologies and general (SEM) modelling details in a conceptualization section unless the contribution is a methodological one), there is again no discussion about the two moderating variables that make part of the authors’ research questions.

10. While the authors discuss details of the TAM model and the estimation of a structural equation model or SEM — and in so doing seem to mix a conceptual model with an econometric one — in the aforementioned section the hypotheses are all stated in terms of the TAM model in general, not in reference to age nor gender and their possibly moderating effects.

11. The authors still suggest that age effects are equivalent to cohort effects. However, typical “age effects” have to do with biological aging, while “cohort effects” stem from common, often cultural, influences that impact in the same way different individuals born around the same time (these influences are further different from “period effects” such as financial crises or a war). The literature on age-period-cohort analysis in sociology, economics, or epidemiology is quite mature and clear about those different effects.

METHODOLOGY AND RESULTS

1. The potentially misleading suggestion referred to above, is clear from the methodology employed in the paper, whereby the authors simply discretize the continuous variable age into 3 categories that are afterwards called “generational cohorts.”

2. The discretization exercise may itself be unfit for a cohort analysis, as the width of the authors’ cohorts is not the same across the three age categories used: 18 years for gen Z, 13 years for gen Y, and, most likely, several decades for gen X+.

3. The authors’ age analysis is in the end an attempt to uncover discontinuities or non-linearities when it comes to the moderating effect of age — again, not only is the moderating effect of age not sufficiently motivated, any specific discontinuities or non-linearities are also not discussed or hypothesized.

4. The composite reliability or Cronbach Alpha analysis does not seem to be satisfactory for the construct PIA - Personal Investment Approach, which appears to be a scale constructed for this study. Would removing any item improve the Cronbach Alpha? The average variance extracted for PEU also appears problematic.

5. As for the variance inflation factors (VIFs) used to assess multicollinearity (Table 4), it should be obvious that they are equal to one when there is only one explanatory variable.

6. Tables 2, 4, and 5 (Table 3 appears to be missing), are redundant and unnecessarily long and could be summarized in one or two sentences.

7. The AIC diagnostics in Table 7 are not so clear cut, as the simpler model with continuous age (and thus the one that uses all the age information without discretizing it) appears to be better for some of the intermediate equations.

8. How do the models compare if other information criteria are used, such as the BIC, etc.?

9. The adjusted R-square Table 10 offers little fit information as it refers to one model only. That is, it does not compare the competing model specifications.

10. The results in Tables 8 and 9 (and in Table 6) are of no use to test whether the authors’ suggestion that there might be moderating effects, is plausible. Those tests are in Table 11. Thus, the length of the paper — and its readability — can be greatly improved.

11. Why are the estimates of PEU —> PU and PIA —> PEU smaller among older survey participants but the estimates of PIA —> PU are larger? Is it because older participants are more interested in investment as retirement becomes closer? (see also the conceptualization point 6 above)

12. The previous point is crucial to position the paper and help the authors re-conceptualize their approach and formulate hypotheses. Whether these results are enough to sustain an entire study / paper is another question.

13. As in any survey not designed to account for endogeneity, all the antecedent variables of BIU may be endogenous. If not the least because BIU might be what causes — within the survey — respondents to answer the way they do to the questions measuring the antecedents (PEU, PU, etc.). Again, this is difficult to account for, but should be discussed as a possible limitation of the study.

14. The use of a highly educated sample, as the authors call it, also limits the generalizability of the results.

Reviewer #3: Thank you for submitting the revised manuscript and addressing previous review comments comprehensively. I have a few more suggestions comments for improving the discussions in the paper.

1. While the gender comparison did not produce notable differences, the paper does describe some minor gender gaps that may still require attention. I suggest that the authors note finding minor gender differences and stating their nature in the abstract, as just saying "we do not find significant differences between male and female" may lead readers to think that no differences were found.

2. Overall, it is interesting - refreshing yet surprising - that no significant gender gaps were found, as much of research on new technology adoption in the recent years show that men are more accepting of new technologies than women in various domains. Is there something that's different about the nature of robo-advisors that may contribute to the lack of differences? Could it be a result of the sample composition, e.g., did males and females in the sample share comparable characteristics, or were there characteristics of the sample that may have contributed to the muted gender effects?

3. The very last paragraph stating limitations of the study needs to be elaborated much further. The sample composition pose additional challenges and limitations than what the authors have mentioned, as there are additional characteristics that make the sample far from representative. A deeper discussion of the sample limitations, and how these may have contributed to the findings need to be discussed. Additionally, it would be helpful to see examples and suggestions around the "behavioral studies" proposed to improve the objectivity and validity of the research.

4. While the paper is written in standard language and does not have major grammatical errors, use of non-academic language and subjectively toned sentences are noticeable throughout the discussions which seem unfit for a research manuscript.

7. PLOS authors have the option to publish the peer review history of their article (what does this mean?). If published, this will include your full peer review and any attached files.

Reviewer #2: No

Reviewer #3: No

---

## [Decision Letter · Decision Letter 2]

18 Apr 2022

PONE-D-21-14526R2Robo-advisor acceptance: do gender and generation matter?PLOS ONE

Dear Dr. FIGA'-TALAMANCA,

Thank you for submitting your manuscript to PLOS ONE. After careful consideration, we feel that it has merit but does not fully meet PLOS ONE’s publication criteria as it currently stands. Therefore, we invite you to submit a revised version of the manuscript that addresses the points raised during the review process.

We look forward to receiving your revised manuscript.

Kind regards,

Ali B. Mahmoud, Ph.D.

Academic Editor

PLOS ONE

Journal Requirements:

Reviewers' comments:

Reviewer's Responses to Questions

**Comments to the Author**

1. If the authors have adequately addressed your comments raised in a previous round of review and you feel that this manuscript is now acceptable for publication, you may indicate that here to bypass the “Comments to the Author” section, enter your conflict of interest statement in the “Confidential to Editor” section, and submit your "Accept" recommendation.

Reviewer #2: (No Response)

Reviewer #3: All comments have been addressed

2. Is the manuscript technically sound, and do the data support the conclusions?

Reviewer #2: Yes

Reviewer #3: Yes

3. Has the statistical analysis been performed appropriately and rigorously? 

Reviewer #2: N/A

Reviewer #3: Yes

4. Have the authors made all data underlying the findings in their manuscript fully available?

Reviewer #2: Yes

Reviewer #3: Yes

5. Is the manuscript presented in an intelligible fashion and written in standard English?

Reviewer #2: Yes

Reviewer #3: Yes

6. Review Comments to the Author

Reviewer #2: I'm pleased that the authors have considered seriously several suggestions to improve their work. Thus, the research problem and specific research questions are now clearer, and as a result so is its contribution. Also, the conceptualization is tighter, namely in respect to the main moderating variable being analyzed, i.e. age, and there are now explicit hypotheses being tested. The paper no longer oversells nor speculates, which makes it more rigorous and increases its validity -- hopefully translating into a larger impact over time.

Having said that, I would like to stress that a generational or cohort analysis is not what the authors do in this paper. Apart from the usually loose definition of generational groups that the authors also mention, the authors' generational groupings do not support their definition. As the authors state, "generations have a different culture" but it is hard to accept that individuals born in say 1995 have the same culture, i.e., belong to the same cohort, as individuals born in say 2010 (some fifteen years later), as it is implied by their grouping into Generation Z.

It is even harder to accept that individuals born in say 1979 have the same culture, i.e., belong to the same cohort, as individuals born in say 1960 (some nineteen years before), as it is implied by their grouping into Generation X+.

I insist this is a fundamental issue in the paper. And I would like to again explain that what the authors really do is to discretize (group, if you will) the age variable they have included in their survey. Apart from issues regarding the loss of information when discretizing a continuous variable that I've discussed in the first round, please note that such discretization, which is arbitrary, cannot by any means be seen as a cohort analysis.

Actually, there are no "generation"-like arguments from say sociology in the conceptualization section nor in the concluding sections.

The analysis really is a typical age moderation analysis and there's nothing wrong with that.

Lines 279 to 285, which by the way are the only ones in the paper offering some rational for why the age dimension may be interesting to look at as a moderator for robo-advice acceptance, are "age"-like arguments rather than "cohort"-like.

Therefore, I would strongly recommend the authors to remove all "generation"-like terminology in the paper (including in the title and in the abstract) and instead stick with the "age"-like terminology.

Lines 146 to 148 can then be re-written as follows: "Respondent ages range from 18 to 67 years. To test our moderation hypotheses, we grouped respondents by birth year as follows: Younger (born...), Middle (born...), Older (born...).

Please note that if there are respondents who were born in 2012 (and that you labeled Generation Z), then in 2019 by the time you have collected the data, they were 7 years old. You mention that the youngest respondents were 18, however.

Methodologically, please note that when reporting structural equation model results it is customary to report a correlation matrix with all variables included in the estimation as well as SEM fit indices, e.g., chi-square test, CFI, RMSEA.

Minor comments:

Table 2 can (should) be reduced in size.

Figures 1 and 2 should have a legend stating that the figures in parentheses are p-values.

Reviewer #3: Thanks for thoroughly reviewing and responding to previous comments. I believe all of suggestions were properly addressed.

7. PLOS authors have the option to publish the peer review history of their article (what does this mean?). If published, this will include your full peer review and any attached files.

Reviewer #2: No

Reviewer #3: No

---

## [Editor Report · Decision Letter 3]

23 May 2022

Robo-advisor acceptance: do gender and generation matter?

PONE-D-21-14526R3

Dear Dr. FIGA'-TALAMANCA,

We’re pleased to inform you that your manuscript has been judged scientifically suitable for publication and will be formally accepted for publication once it meets all outstanding technical requirements.

Kind regards,

Ali B. Mahmoud, Ph.D.

Academic Editor

PLOS ONE
---

## [Editor Report · Acceptance letter]

3 Jun 2022

PONE-D-21-14526R3 

Robo-advisor acceptance: do gender and generation matter? 

Dear Dr. Figà-Talamanca:

I'm pleased to inform you that your manuscript has been deemed suitable for publication in PLOS ONE. Congratulations! Your manuscript is now with our production department. 

Kind regards, 

on behalf of

Dr. Ali B. Mahmoud 

Academic Editor

PLOS ONE